# Crop Wild Relatives: A Valuable Source of Tolerance to Various Abiotic Stresses

**DOI:** 10.3390/plants12020328

**Published:** 2023-01-10

**Authors:** Aliki Kapazoglou, Maria Gerakari, Efstathia Lazaridi, Konstantina Kleftogianni, Efi Sarri, Eleni Tani, Penelope J. Bebeli

**Affiliations:** 1Institute of Olive Tree, Subtropical Crops and Viticulture (IOSV), Department of Vitis, Hellenic Agricultural Organization-Dimitra (ELGO-Dimitra), Sofokli Venizelou 1, Lykovrysi, 14123 Athens, Greece; 2Laboratory of Plant Breeding and Biometry, Department of Crop Science, Agricultural University of Athens, Iera Odos 75, 11855 Athens, Greece

**Keywords:** adaptation, alfalfa, breeding, genetic resources, grain legumes, tomato, woody perennial crops

## Abstract

Global climate change is one of the major constraints limiting plant growth, production, and sustainability worldwide. Moreover, breeding efforts in the past years have focused on improving certain favorable crop traits, leading to genetic bottlenecks. The use of crop wild relatives (CWRs) to expand genetic diversity and improve crop adaptability seems to be a promising and sustainable approach for crop improvement in the context of the ongoing climate challenges. In this review, we present the progress that has been achieved towards CWRs exploitation for enhanced resilience against major abiotic stressors (e.g., water deficiency, increased salinity, and extreme temperatures) in crops of high nutritional and economic value, such as tomato, legumes, and several woody perennial crops. The advances in -omics technologies have facilitated the elucidation of the molecular mechanisms that may underlie abiotic stress tolerance. Comparative analyses of whole genome sequencing (WGS) and transcriptomic profiling (RNA-seq) data between crops and their wild relative counterparts have unraveled important information with respect to the molecular basis of tolerance to abiotic stressors. These studies have uncovered genomic regions, specific stress-responsive genes, gene networks, and biochemical pathways associated with resilience to adverse conditions, such as heat, cold, drought, and salinity, and provide useful tools for the development of molecular markers to be used in breeding programs. CWRs constitute a highly valuable resource of genetic diversity, and by exploiting the full potential of this extended allele pool, new traits conferring abiotic-stress tolerance may be introgressed into cultivated varieties leading to superior and resilient genotypes. Future breeding programs may greatly benefit from CWRs utilization for overcoming crop production challenges arising from extreme environmental conditions.

## 1. Introduction

Despite the fact that global hunger rates were static since 2015, they increased rapidly from 2019 onwards, and humanity is facing again a rising global hunger crisis [1]. At the same time, the world population is growing and is expected to reach 9.7 billion in 2050 and 10.4 billion in 2100 [2]. Human activities, such as poor management practices, changes in diet preferences, increasing competition for land, water and energy use, soil degradation, crop diseases, and climate change, are some of the factors that challenge food productivity [3]. Climate change with a global average temperature increase of at least 1 °C since the industrial revolution coupled with an increase in the frequency and intensity of extreme weather events, such as droughts, floods, heat waves, and storms, endangers agri-food systems worldwide [4].

At present, agricultural land and global production are sufficient to feed the world population, although economic and social inequalities and distribution difficulties leave a significant part of the population in a state of starvation. However, demand for food is expected to outstrip the production capacity of current agricultural systems as an increase of 70% in agricultural yields is expected to be needed to feed the population in 2050 [3]. Therefore, according to FAO, we are facing the most dangerous period for agriculture in human history [4].

In the last decades, cultivated land has increased worldwide by about 12% at the expense of natural ecosystems. This implies destruction of forests and serious impacts on wild biodiversity. Therefore, the percentages of land converted to agricultural land should be kept as low as possible. Other solutions, such as improving agricultural practices, creating new environmental policies, changes in diet, and reducing food waste, are decent but partial [3,5].

Regarding modern high-yielding improved varieties, it is reported their high productivity has been achieved through a simultaneous reduction of their genetic base [6]. These varieties were created to meet ideal field conditions with an adequacy of inputs, such as water and fertilizer [7,8]. Furthermore, these varieties tend to emphasize reproduction rather than defense and competition mechanisms. These are major problems nowadays as agriculture is threatened by extreme conditions [9].

To comprehensively address the problems of modern agriculture, it is therefore crucial to breed novel crop varieties resistant or tolerant to environmental stresses [10]. Efforts should be made, focusing on traits that could be introduced to face key abiotic plant stresses, such as drought, salinity, and extreme temperatures. In this aspect, scientists should exploit all the available genetic diversity [5]. However, there are obstacles because of the gradual loss of alleles and genetic bottlenecks due to the thus far plant breeding efforts, plant domestication, and the extinction of plant species. Pointedly, both plant breeding and domestication rely on human selection, and all forms of selection lead to a loss of genetic variability since only genotypes that are superior for certain traits are advanced [11].

The solution to recover lost diversity, overcome breeding bottlenecks, and avoid genetic vulnerability is to expand the existing gene pools of cultivated plants. One possible strategy to achieve this objective is to utilize sources of wild desirable genes, namely, crop wild relatives (CWRs) [12]. Crop wild relatives (CWRs) are ancestors or progenitors of domesticated crop species as well as other close relatives throughout evolutionary history [12] that can naturally cross successfully, sometimes implicating assisting methods, with cultivated species. CWRs are taxonomically related to domesticated plants and may belong to the same species. However, they exist as wild species in natural habitats in and near their centers of origin [13]. Notably, they are widespread on all continents besides Antarctica, and several can be found in the Vavilov’s diversity centers and their adjacent regions [10].

Wild relatives, unlike domesticated species, were not subjected to strict anthropogenic selection pressure with an emphasis on traits related to plant yield under optimal controlled conditions [14]. Instead, throughout evolutionary history they were exposed and adapted in an abundance of adverse environments and kept evolving to adjust and survive under such harsh conditions [9]. Crop wild relatives possess a plethora of genes that confer increased resistance to abiotic stresses [14] and represent a source of alleles that are absent from modern cultivars that have significant agronomic value. They also feature higher genetic and phenotypic variability than domesticated species and thus provide breeders with a rich gene pool which constitutes a useful genetic resource for breeding programs. Moreover, this resource is likely to broaden the genetic base of cultivated varieties by introducing economically important genes, critical for meeting the challenges of food crisis and climate change [13].

In the present review, we gathered information regarding the utilization of the CWRs as a source of important abiotic traits for some representatives of crops species with high economic value apart from cereals. The first representative is tomato (*Solanum lycopersicum* L.), one of the most extensively studied vegetable species with great importance for human nutrition. The second category includes (a) *Medicago sativa* L., which is a fodder legume with high nutritional value, used intensively for animal feed and (b) representatives of grain legumes, such as cowpea (*Vigna unguiculata* (L.) Walp.) and peanut (*Arachis hypogea* L.), that are often cultivated in marginal lands and confront unfavorable environmental conditions [15]. The last category includes representatives of woody and perennial crops that have been using crop wild relatives not only for crosses but also as rootstocks for grafting. Although this review focuses on abiotic stressors, for this last category, some CWRs that played a crucial role in the history and the creation of the modern varieties with tolerance to biotic stresses are also mentioned.

## 2. The Exploitation of the CWRs in Specific Cultivated Species

### 2.1. Tomato

Cultivated tomato (*Solanum lycopersicum*) is one of the most economically important crops also used as a model crop for vegetables [16]. It is a diploid species with a haploid set of 12 chromosomes and genome size of approximately 950 Mb [17]. There is a plethora of tomato wild relatives with specific traits, including *Solanum pimpinellifolium, Solanum nigrum, Solanum pennelli, Solanum peruvianum,* and *Solanum chilense* [18] among others, which have shown tolerance to different abiotic stresses and adaptation mechanisms to different and extreme environmental conditions. Indicatively, *Solanum chilense* can grow in the desert due to its long primary roots and its extensive secondary root system. Moreover, it has been proved that *Solanum pennellii* utilizes water availability in soil efficiently under drought conditions, while *Solanum cheesmanii* and *Solanum peruvianum* can grow in salty coastal areas due to different adaptation mechanisms they have developed on their root systems [19]. The exploitation of such huge genetic diversity existing in CWRs in tomato breeding efforts for abiotic stress tolerance can provide new varieties with an enormous reservoir of adaptive traits. Several examples of tomato CWRs regarding their tolerance to abiotic stressing factors are mentioned below and reported extensively in Table 1.

#### 2.1.1. Drought Stress Tolerance

Drought is an important limiting factor of crop production, especially in the context of global climate change. Several drought-tolerant (quantitative trait loci) QTLs or genes have been identified in tomato CWRs, but they have not been proven as successful as expected [66]. The development of advanced backcross introgression lines (BILs) provides a useful alternative method for the transfer of drought-tolerant genes [10]. Eshed and Zamir [67] presented a novel population consisting of 50 introgression lines originating from a cross between the green-fruited species, *Lycopersicum pennellii,* and the cultivated tomato (cv M82). Since then, many researchers used these lines in breeding programs to investigate promising genes and QTLs for drought stress among other stress factors, concluding that despite the difficulties, this approach may be the best strategy if no other effective breeding alternatives are available [10,68,69]. In fine-mapping drought tolerance within several introgression of *S. pennellii* and the parental line cv M82, cleaved amplified polymorphic sequences (CAPS) markers have been developed and screenings for root morphological traits performed to identify plants putatively inheriting a root architecture compatible with drought tolerance [68].

Advances in genetics and genomics have improved the understanding of structural and functional aspects of the tomato genome [69]. Moreover, genes with high homology to *FQR1-like* NAD(P)H dehydrogenase, known for its antioxidant properties, were also identified *in S. pimpinellifolium* [70]. Further study of these unique orthologues might give insight into the adaptation of *S. sitiens*, another drought and salt-tolerant tomato CWR in water-limited environments [26]. In the future, new technologies, such as CRISPR-Cas9, could enable the transfer of genes underlying drought tolerance from tomato CWRs to cultivated tomato varieties [23].

#### 2.1.2. Salt Stress Tolerance

Tomato is susceptible to salt stress, which leads to substantial productivity reduction. CWRs, such as *S. pimpinellifolium, S. pennellii*, and *S. chilense*, can tolerate these adverse conditions of salinity. Recently, the genome of *S. pimpinellifolium* has been sequenced [70] and revealed an interesting finding for genes associated with abiotic stress, such as salinity. Additionally, further study of some genes from *S. chilense* associated to abiotic stress [71] can be a starting point to annotate new genes and the applicability for a genome-wide analysis. Salinas-Cornejo et.al. [72] provided information that expression levels of *SlAREB1*, a member of the abscisic acid-responsive element binding protein (AREB), are correlated with the degree of drought and salt tolerance presented by transgenic tomato plants. They also identified an important number of genes regulated by SlAREB1 protein and associated with both abiotic and biotic stress responses.

Furthermore, many QTLs have been mapped for salt stress tolerance during the different growth stages in tomato, For instance, (a) at the seed germination stage, indicating *S. pennellii* and *S. pimpinellifolium* as potential tolerant sources of salt tolerance with several QTLs mapped [27,73], and (b) at the vegetative and reproductive growth stages with regard to fruit number, fruit weight, and fruit yield, several QTLs from *S. pimpinellifolium* [74,75] have been annotated. The reported QTLs and major associated genes would be possibly transferred to suitable genetic backgrounds of cultivated tomato genotypes to develop tolerant cultivars against salt stress [76].

#### 2.1.3. Heat/Cold Stress Tolerance

Cold stress reduces uptake of water and nutrients in plants, leading to nutrient starvation within cells. Furthermore, heat stress leads to marked alterations in the physiology and metabolism of plants [77]. On the other hand, the characterization of important gene families and their relative expression under low and high temperatures have been reported for a wide variety of plants [78,79,80]. Studies focusing on tomato chilling responses revealed over-expression of the chloroplast gene family encoding heat shock proteins (HSPs) and concomitant reduction in ROS and lipid peroxidation, reflecting an increase in photosynthetic performance [81]. Tolerance mechanisms utilized by plants against chilling conditions involve the increased expression of genes that reduced the intensity of oxidative damage induced by cold stress [82]. Catalase, a crucial ROS-scavenging enzyme, eliminates hydrogen peroxide in the cell cytoplasm and contributes to the scavenging of H_2_O_2_ [83]. The seed priming-induced method increases CAT activities for several species studied, including tomato [81]. Further investigation into tomato CWRs for more genes that induce tolerance mechanisms for cold and heat stress and their exploitation in cultivated tomato breeding programs is a promising approach for solving problems caused by extremely low/high temperatures.

### 2.2. Alfalfa

Legume plants make up one-third of the world’s major agricultural yield and are significant sources for human and animal consumption [84]. *Medicago sativa* (alfalfa), the most significant legume fodder, has been cultivated in more than 80 countries with a total surface area of 32 million hectares available globally [85]. Alfalfa is regularly exposed to harsh environments in the major regions of the world, including drought in Argentina and northern China, cold temperatures in Russia and Canada, and saline/alkaline soils in California, America, and Australia. Environmental stress in these places has had a significant impact on alfalfa productivity and quality [86,87,88,89]. Long-term domestication of cultivated alfalfa may have resulted in decreased tolerance for severe abiotic and biotic stressors because of the emphasis placed on features linked to high production. The key to accelerating *M. sativa’s* breeding is the use of genetic variants underlying agronomic features in wild relatives close to the cultivated Medicago [90]. Therefore, genomic data from wild species might offer important insights for enhancing features linked to the adaptability to stressful situations in legume forages. For breeding alfalfa varieties with great tolerance to environmental challenges, genetic resources rich in alleles adaptable to severe environments are highly needed.

The genus Medicago includes wild species that are closely related to *M. sativa*, including *Medicago truncatula* (a model plant for legumes) [34], *Medicago ruthenica* [36], *Medicago polymorpha* [91], and *Medicago falcata* [38]. Ιn the present study, emphasis was given to one of the most promising wild species. *Medicago ruthenica* (L.) Trautv. is a natural grassland plant that is widely distributed in hillsides, mixed grass steppes, and meadows in Siberia, Mongolia, and northern China [92]. It is an allogamous diploid (2n = 2x = 16) perennial legume fodder with a re-assembled genome of 904.13 Mb [93]. It is very closely related to alfalfa [93]. Long, chilly winters and dry, saline soils limit *M. ruthenica’s* distribution area [94]. As a result, *M. ruthenica* must have developed powerful defenses to withstand the harsh conditions, such as drought, subfreezing temperatures, and saline soil. *M. ruthenica* is thought to be a rather uncommon species among Medicago species that is highly adapted to stressful conditions, and whose prospective applicability is favorably assessed in low-input environments. The roles of differentially expressed abiotic stress-related genes, such as the *AP2/ERF* family, *MYB/MYB-related* family, *bZIP, bHLH,* and *WRKY* from *M. ruthenica* in conferring stress tolerance have not been fully elucidated [95,96]. These genes play important roles in many different regulation mechanisms of diverse abiotic stresses. Numerous studies have demonstrated the transcription factors, *bZIP*, *WRKY,* and *AP2/ERF,* have a role in the transduction of the ABA signal and stress responses in plants through their interaction [97]. 

#### 2.2.1. Drought Stress Tolerance

In addition to being a close relative of alfalfa, *M. ruthenica* is a perennial species with a similar genome size, life cycle, and pollination system. More importantly, because it is a wild species with numerous accessions that is found widely in arid and/or semi-arid areas, it is highly tolerant to drought stress. As a result, it has been used as parental material to breed alfalfa cultivars that are tolerant to environmental stress, which has improved alfalfa tolerance to adverse environments [35,93].

*M. ruthenica’s* drought tolerance was compared to that of *M. truncatula, M. varia, M. falcata*, and two cultivars of alfalfa by Wang et al. [36]. Among the tested legume species, *M. ruthenica* seedlings showed the greatest resilience to drought stress. The strongest resistance of *M. ruthenica* to drought stress among the examined legume forages was demonstrated by the fact that while exposure to drought significantly reduced the survival rates of other legume forages, the same treatment had little impact on the survival rates of *M. ruthenica* seedlings. Additionally, it was found *M. ruthenica* and *M. truncatula*, respectively, have 37 and 23 of the *AP2/ERF* family, drought-responsive *TF* genes. Twenty-one and ten drought-responsive *TFs* from the *MYB/MYB-related* family in *M. ruthenica* and *M. truncatula*, respectively, were discovered [35]. The knowledge of *M. ruthenica’s* genome and the discovery of its resistance genes can help to improve agronomic traits linked to high yield and exceptional tolerance to environmental stress using a molecular breeding strategy.

#### 2.2.2. Salt Stress Tolerance

Abiotic stresses, such as salt stress, have an impact on plant productivity and growth. *Medicago ruthenica* exhibits exceptional stress resistance, making it a valuable gene resource for enhancing other plants’ stress tolerance. Two differentially expressed genes *(DEGs) (MrERF, MrbZIP*) from *M. ruthenica* have not yet been thoroughly characterized in terms of their functions in salt tolerance. Wu et al. [95] demonstrated the transgenic lines of tobacco over-expressing these genes grew more successfully than wild types exhibiting greater height, more branches, and earlier flowering. Additionally, compared to wild type tobacco, the seed yield of transgenic tobacco was considerably higher. Furthermore, it was demonstrated that *MrERF* or *MrbZIP* may be rapidly expressed in leaves by NaCl since three transgenic tobacco lines outgrew the wild one in terms of leaf growth, with *MrERF* and *MrbZIP* having the best growth. Thus, *MrERF* and *MrbZIP* can increase the germination rate of Medicago under salt stress because they greatly increased the germination rate of transgenic tobacco lines. Plant height, biomass, and the root-to-shoot ratio of transgenic tobacco expressing *MrERF* or *MrbZIP* all significantly increased when exposed to NaCl. Lastly, under salt stress, *MrERF* and *MrbZIP* transgenic lines’ roots length grew by approximately 2.03 and 2.19 fold of wild type, respectively [96].

#### 2.2.3. Cold Stress Tolerance

*M. ruthenica* is also an important resource for the genetic improvement of alfalfa in response to cold stress. Shu et al. [94] performed an RNA-Seq analysis of the *M. ruthenica* transcriptome in response to cold stress using high-throughput nucleotide sequencing. A total of 894 genes were identified that responded to cold stress. Expressions of *MrUN10866, MrUN33504, MrUN37588*, and *MrUN40182* were induced by cold stress [95]. Numerous transcription factors (TFs) have been identified, and they all play crucial roles in how plants react to abiotic stressors, such as cold, including AP2/ERF, bHLH, MYB, WRKY, C2H2, and NAC. The AP2/ERF TF family, whose members have been extensively characterized for their involvement in cold tolerance, was the largest. This finding suggests these genes play a crucial role in the abiotic stress response, and they may be utilized in breeding alfalfa [95]. The wild Medicago species that presented tolerance in the above-mentioned abiotic stresses are presented in Table 1.

### 2.3. Grain Legumes

Grain legumes are often cultivated in marginal lands and confront unfavorable environmental conditions [15] that prevail in these areas. Typically, marginal areas are characterized by poor soil fertility and are usually prone to abiotic stresses, such as drought and salinity [9]. Breeding for tolerant grain legume genotypes to various abiotic stresses is therefore of primary importance, especially given the impending climate change. In this review, we focused on progress regarding CWRs assessment and implementation in breeding of two summer grain legume species, namely cowpea (*Vigna unguiculata* (L.) Walp.) and groundnut (*Arachis hypogaea* L.), as the climate change effect becomes more and more apparent in Europe; thus, they consist of crops with increasing rates of cultivation in the area [97].

#### 2.3.1. Cowpea

Cowpea (*Vigna unguiculata* (L.) Walp.) (2n = 2x = 22) is a primarily autogamous, annual, grain legume domesticated in Africa in two parallel primary centers of origin [98,99,100,101]. It belongs to the genus *Vigna* which comprises about two hundred different species [102], among them over one hundred wild species [45,103,104]. A plethora of endemic cowpea wild types are still present in Africa [105] that are considered progenitors of cultivated cowpea [98] and contain valuable genetic material for breeding. Recently, African cowpea genome size was revised and estimated through cytometric studies on 640.6 Mb [104], while for asparagus bean (*Vigna unguiculata* ssp. *sesquipedalis*) a 632.8 Mb genome size is reported [105]. The difference between the two species genome size is mainly because of changes in the Gypsy retrotransposons contained [104].

Genes of the *NAC* family (*VuNAC1, VuNAC2*) [106], *WRKY* family (*VuWRKY*) [107], *DREB* family (*VuDREB 2A*) [108], *pUCPs* family (*VuUCP1a, VuUCP1b*) [109], *Aox* family (*Vu Aox1, Vu Aox2*) [110], *HSP* family (*VuHSP17*.7) [111,112], and *LEA* family [113] have been found among others to be differentially expressed in cowpea under abiotic stresses conditions, promoting abiotic stress tolerance.

##### Drought and Other Types of Stress Tolerance

Cultivated cowpea types have proved more prone to drought than the wild *Vigna* material [114]. High levels of drought tolerance have been reported for *V. heterophylla* and *V. kirkii*, while high-temperature occurrence tolerance have been reported in *V. hainiana* and *V. stipulacea* [40]. *V. exilis, V. trilobata,* and *V. riukiensis* have also been characterized as drought tolerant [39] as they express genes related to ABA biosynthesis and proline biosynthesis. Antioxidant capacity, accommodation of small leaves that increase the heat flux from the leaf surface, and hairiness of leaves consist of mechanisms for drought and heat tolerance of wild *Vigna* species [114]. *Vigna riukiuensis* is especially characterized by a deep and wide root system and small size of leaves that renders it a heat-tolerant species [43,44,45]. Wild *Vigna* germplasm materials, such as *V. minima* and *V. indica* [115], have also been found to be tolerant to acidic and limestone type of soils, and *V. vexillata* has been found to be a water-logging-tolerant wild species [116]. *Vigna* wild species tolerance to abiotic stresses is presented in Table 1.

##### Salt Stress Tolerance

Salinity tolerance has been reported for *V. luteola, V. marina, V. nakashimae*, *V. vexillata var. macrosperma, V. riukiuensis,* and *V. trilobata* [41,42,43,44]. Salt tolerance mechanisms of wild cowpea species include Na+ exclusion, increased antioxidant ability, osmotic regulation, and changes in hydraulic conductance [117]. Accessions of wild species of *V. nakashimae* and *V. riukiuensis* express salt tolerance through Na+ filtration by roots and stems to prevent uptake into leaves and accumulation of large Na+ amount throughout the whole plant, respectively [43]. Through screening of Asian *Vigna* wild types, Naito et al. [39] led to the identification of genes related to salt stress response, such as sodium and potassium transporters, while salt-tolerant species presented active transcription of *SOS1* and *SOS2* genes. They also found genes related to salt stress that are related to ABA biosynthesis [39]. *Vigna marina* was found actively to transcribe sodium transporters and antiporters (*NHX1* and *NHD1*), while *NHX2* and *HKT1* potassium transporters were transcribed by *V. riukiuensis* [39], leading to increased salt tolerance. Furthermore, lower base water potential of seeds compared with other *Vigna* species renders its seeds able to germinate in soils with increased salt content [42].

Finally, QTLs for salt tolerance were found in *V. marina* ssp. *oblonga*, which is a salt-tolerant type, that could be further used in introducing salt tolerance in cultivated cowpea [41]. However, attempts to cross the cultivated African cowpea with salt-tolerant [117] *V. vexillata*, the closest species to cowpea [118], have so far proved fruitless [119,120] despite the creation and characterization of an interspecific hybrid created by Gomathinayagam et al. [121]. As wild *Vigna* relatives are interesting material for the improvement of the cultivated *Vigna* [46,122], the continuation of the crossing effort is considered critical. Wild cowpea germplasm variation should be more intensively exploited [123] as it consists of a valuable source of abiotic stresses tolerance [124].

#### 2.3.2. Groundnut

Groundnut (*Arachis hypogaea* L.) or peanut (2n = 4x = 40) (AABB) is an annual grain legume and oil crop with a primarily self-pollinated mating system [125]. As it was formed after a hybridization between diploid species, *A. duranensis* (AA) and *A. ipaensis* (BB), followed by chromosome doubling [126], it has a wide gene pool [122,125]. Cultivated groundnut genome size is approximately 2.7 Gb, very similar to the sum of its two wild progenitors, *A. duranensis* (1.25 Gb) and *A. ipaensis* (1.56 Gb) [127]. Crop wild relatives of groundnut constitute a valuable but underutilized genetic source due to difficulties in introgression of genes into the cultivated species, owing to ploidy discrepancy among groundnut (allotetraploid species) and the wild relatives (mainly diploid species) [125,128] as well as sterility barriers [129]. Drought and high temperatures constitute the major abiotic stress factors for groundnut, whereas salinity tolerance is important in many areas [125].

Plant heat shock factors (HSFs) play a key role in groundnut response to various environmental stresses by regulating the expression of stress responsive genes, such as heat shock proteins (HSPs), dehydration responsive element-binding proteins (DREBs), late embryogenesis abundant proteins (LEA), and abscisic acid response element-binding proteins AREB [130]. CRT element-binding factors (CBFs) were also found to respond to various plant stresses [131,132,133,134]. 

Dehydration responsive element-binding protein (DREB) genes are reported to increase transpiration efficiency under water-limiting conditions (*AtDREB1A, DREB1A*) [135,136,137]. A DREB factor, namely *PNDREB1,* was also identified by Zhang et al. [138] to respond to low temperatures and osmotic stress. However, no response of this gene to salinity has been observed. An ABA synthesis gene, *AhNCED1,* and three dehydration-induced transcription factor genes were also found to be differentially regulated in groundnut under drought conditions [139].

Ethylene-responsive factor (ERF) regulates gene expression associated with abiotic stress tolerance [140] through the activation of ABA [141]. In groundnut, ERF genes were found to be induced after the application of abiotic stresses, such as drought, cold, heat, and salinity [140]. Additionally, *AhLea-3* has been reported to be related to salt tolerance [142]. *AtNHX1*, a vacuolar Na+/H+ antiporter in *Arabidopsis thaliana*, mediates the transport of Na+ and K+ into the vacuole, enhancing salt tolerance [143]. Overexpression of the *AtNHX1* gene was also found to improve salt and drought tolerance in transgenic groundnut [144]. Moreover, stress-inducible expression of *AtHDG11* in transgenic peanut lines resulted in up-regulation of various stress responsive genes (*LEA, HSP70, Cu/Zn SOD, APX, P5CS, NCED1, RRS5, ERF1, NAC4, MIPS, Aquaporin, TIP, ELIP*) leading to improved drought and salt tolerance [145].

##### Drought Stress Tolerance

Wild groundnut species genes that are drought stress involved are discriminated into two groups: (i) genes that are plant cell protectors and act as upstream regulators and (ii) regulatory genes, such as transcription factors, implicated in the ABA pathways [146]. Tolerance to abiotic stresses has been recorded in *A. stenosperma* [128], while Guimarães et al. [147] identified a remarkable number of transcription factors and genes related to drought stress in a peanut ancestor, *A. duranencis*. In the same wild species, a great number of differentially expressed genes were recorded under drought stress treatment [148]. *Arachis duranensis* tolerance to drought is based on restricted plant transpiration behavior upon stress implementation. Several characteristics associated with drought response were detected in *A. dardani*, such as leaf angle adjustment [48], and in *A. duranensis* high photosynthetic rate, stomatal conductance, transpiration rate, lower leaf temperature, and vapor pressure [50]. Drought-responsive candidate genes, such as *Expansin, Nitrilase, NAC*, and *bZIP* transcription factors, displaying significant levels of differential expression during stress application in *A. duranensis* and *A. magna* were identified, while they possess drought response mechanisms, including signal transduction, primary metabolism, hormone homeostasis, and protection of cellular structures [53]. More recently, genes encoding the drought-responding fatty acid, desaturase, were also identified in groundnut progenitors and presented to be homologous to peanut [149].

##### Heat/Cold Stress Tolerance

A heat-tolerant genotype of the wild species *A. glabrata* (*A. glabrata* 11824) and a cold tolerant genotype of *A. paraguariensis (A. paraguariensis* 120142) were identified by [55] through screening of thirty-six different wild genotypes. A total number of sixteen and seventeen heat shock transcription factors were found in *A. duranensis* and *A. ipaensis*, respectively, that are commonly known to protect plants from abiotic stresses [54]. Non-specific lipid transfer proteins (nsLTPs) that are known to transfer various lipid molecules between lipid bilayers in plants were identified in *A. duranensis* that respond to salinity and low-temperature conditions [51]. Tolerance assessed in *Arachis* wild species is presented in Table 1.

##### Combined Abiotic Stress Tolerance

Genes and transcription factors identified in *Arachis* wild species were in many cases responding to a complex of abiotic stresses. In wild groundnut *A. diogoi,* the expression of gene *AdDjSKI* was induced under heat, salinity, drought, and osmotic stresses and seems to be related to the photosynthetic mechanism of plants [49]. Recently, NAC transcription factor genes that are implicated in salt and drought responses of many plant species were identified in *A. hypogaea* as well its progenitors [56]. Valine-glutamine sequences that are related to environmental changes were also assessed in wild species, *A. duranensis, A. ipaensis,* and *A. monticola*. Genes of the *mTERF* family that mediate acclimation of plants to adverse environmental conditions were also identified in *A. duranensis* and *A. ipaensis* that were distributed over their ten chromosomes [150]. Transcriptomic analyses on the wild and highly adaptable *A. glabrata* revealed a plethora of transcript factors to be expressed under drought, salt, and cold stress implementation [52]. Finally, a great number (4513) of differentiated expressed genes (DEGs) was also recorded to be expressed under UV exposure and dehydration in *A. stenosperma* by Martins et al. [57], mainly associated with cell signaling, protein dynamics, hormonal and transcriptional regulation, and secondary metabolic pathways. These genomic findings provide useful tools for the further improvement of the species in abiotic stresses.

Synthetic amphidiploid and autotetraploid groundnuts were created to overcome genetic barriers of groundnut breeding [151]. Synthetic groundnut germplasm was mainly screened for biotic stress tolerance [152,153,154,155,156] and introgressed into cultivated material [157,158]. The variability that synthetics often express could possess hidden alleviation for abiotic stress tolerance that remains mostly unexploited. Stress tolerance of the tetraploid groundnut species is not always expressed in the same way as their wild diploid relatives [154]. Bera et al. [159] found two interspecific derivatives, (NRCGCS-296 (J11 *x A. duranensis*)) and (NRCGCS-241 (GG 2 *x A. cardenasii*)), that presented high germination tolerant index and promptness index while applying 250 mM NaCl and therefore were characterized as tolerant to salinity. *WRKY* and Na+/H+ genes were also assessed as responsible for inducing tolerance in the synthetic hybrids.

### 2.4. Woody Perennial Crops

Climate changes leading to temperature alterations (increased heat or cold), extreme weather phenomena (e.g., dry spells, heat waves, heavy rainfalls), and water availability (drought or flooding conditions) pose threats to the cultivation of woody perennial crops. Changes in weather patterns subsequently exacerbate biotic stresses and the spread of diseases. Ultimately, these adverse abiotic and biotic effects significantly compromise yield and quality of the final product. In recent years, numerous efforts have been undertaken to expand the genetic pool of woody perennial crops by exploiting the genetic diversity of wild relatives and introgressing new desirable climate-resilient traits into cultivated varieties [160,161].

#### 2.4.1. Apple

The cultivated apple, *Malus domestica* Borkh., is a diploid or triploid species with a haploid set of 17 chromosomes and a genome size of approximately 600 Mb [162]. Apple cultivation and production constitute one of the major fruit-producing industries addressing markets worldwide. However, climatic changes introduce a series of environmental stressors which challenge apple yield and fruit quality. Apple breeding could greatly benefit from apple wild relatives to face the challenge from adverse environmental conditions and biotic and abiotic stressors [161]. For example, wild relatives, *Malus floribunda*, *Malus baccata,* and *Malus micromalus,* have been used to pyramid apple scab and powdery mildew resistance genes into progeny [163]. Assessment of a broad range of wild Malus germplasm over the last 30 years has revealed ample potential sources of resistance to a multitude of diseases. Similarly, apple wild relatives in Malus collections have been evaluated and shown to possess traits related to fruit quality as well as abiotic stress resilience, such as cold hardiness and drought tolerance [59]. In addition, investigations on the molecular basis of stress tolerance have indicated a key role for a DREB2 (dehydration-responsive element-binding factor 2) homologue in response to drought, cold, and heat in two highly drought-tolerant wild apple relatives, *Malus sieversii* and *Malus prunifolia* [60,61]. Likewise, *Diacylglycerol kinase* (*DGK*) genes were found to exhibit marked upregulation in response to drought and salt stress in *M. prunifolia* [58]. Comparative analyses between two widely used apple rootstocks (*M. sieversii* and R3) under water deficit conditions demonstrated that *M. sieversii* is more tolerant to drought. Transcriptomic analysis of root tissue showed differential expression of stress-responsive genes associated with oxidative stress, signaling pathways in hormone biosynthesis, and transcriptional regulation between the two genotypes, suggesting these genes play a crucial role in root processes that provide drought tolerance [164]. Moreover, the deciphering of the cold-tolerant wild apple *Malus baccata* genome identified cold-responsive genes (*COR*) that will be useful in marker-assisted selection in breeding programs [62]. Collectively, the Malus wild relatives provide an important genetic resource for incorporating resilience in cultivated apple varieties.

#### 2.4.2. Cranberry

Research focusing on wild cranberry is another example of targeted use of genetic variation in perennial wild populations toward the benefit of breeding resilient varieties [63,165]. Cranberry (*Vaccinium macrocarpon* Ait.), a fruit crop of high economic value in North America, Northern Europe, and Asia, often encounters a series of abiotic and biotic challenges, such as frost damage, high temperatures, drought, flooding, and fungal diseases, which lead to severe production losses. Recently, a collection of many wild cranberry accessions from the northern U.S. and Canada was assessed through environmental association analysis and revealed genomic regions linked to potential abiotic stress tolerance. One hundred twenty-six significant associations between SNP marker loci (many of which tagged genes with functional annotations) and environmental variables of temperature, precipitation, and soil attributes were uncovered [166].

#### 2.4.3. Grapevine

Although the *Vitis* genus is composed of 60 species, the species used predominately for grapevine cultivation is *Vitis vinifera* L. Nevertheless, wild *Vitis* relatives exhibit important traits not found in *V. vinifera,* such as resistance to the devastating ‘Pierce’s disease’ (PD) caused by the bacterium *Xyllela fastidiosa.* Breeding programs focused on a PD-resistant grapevine wild relative, *Vitis arizonica*, to generate PD-resistant lines. Over the years, using *V. arizonica x V. vinifera* crosses, repeated backcrosses with *V. vinifera* and marker-assisted selection (MAS) techniques, breeders managed to develop breeding grapevine lines with PD resistance and 97% *V. vinifera* ancestry [167]. Similarly, to confront two major grapevine fungal diseases, downy mildew (*Plasmopara viticola*) and powdery mildew (*Erysiphe necator*), the wild relative *Muscadinia rotundifolia* was utilized. Crosses with *Vitis vinifera* and subsequent crosses with other *Vitis* hybrids resulted in progeny containing genes implicated in resistance to both powdery and downy mildew [168,169].

On the other hand, molecular studies have begun to elucidate the genetic basis of abiotic stress tolerance displayed by wild *Vitis* relatives. Overexpression of a stress-related gene from the Chinese wild grape *Vitis yeshanensis* encoding a universal stress protein, VyUSPA3, was shown to confer drought tolerance to transgenic *V. vinifera* cv. ‘Thompson Seedless’ [64] (Table 1). In addition, comparative transcriptomic analysis performed between a coastline wild grapevine accession (*Vitis vinifera* L. ssp. sylvestris) which is tolerant to high-salinity levels and the commercial rootstock, Richter 110, a salt-sensitive cultivar, revealed differential gene expression profiles upon salinity stress [65] (Table 1). These findings facilitate the investigation of gene pathways that play key roles in survival under stress conditions and highlight the potential of such grapevine wild relatives as breeding material both for scion and rootstock improvement.

In view of the gloomy projections of 56 to 73% loss of suitable land for viticulture in major wine-producing regions by 2050 [170,171], studies have been focusing on grapevine wild relatives with resilience to climate risk [172]. Recently, associations of wild species SNPs (single nucleotide polymorphisms) with bioclimatic variables and putative adaptation to biotic and abiotic stressors have been explored [173]. In addition, by integrating species distribution models, adaptive genetic variation, genomic load and phenotype, Aguirre-Liguori et al. [174] predicted that certain accessions of the wild grapevine species, *Vitis mustangensis,* are well-suited for future climates and can contribute to grapevine bioclimatic adaptation.

Importantly, commercial rootstocks currently used globally for grapevine grafting were derived from North American wild *Vitis* species. These rootstocks have been used since the second half of the nineteenth century to save European grapevines from the plague of the soil-borne aphid, phylloxera (*Daktulosphaira vitifoliae*) [175]. Moreover, depending on the rootstock, they confer drought and cold tolerance as well as disease resistance to grafted grapevine [176]. Likewise, rootstocks have been used widely for improving other cultivated woody perennials (apple, pear, peach, mango, citrus, etc.). However, in general, relatively few rootstock genotypes are employed in grafting of woody perennial crops. Wild relatives could serve as a significant allele pool for developing new rootstock varieties with advantageous traits that would impart the grafted plant with resilience to environmental stressors [160].

## 3. Conclusions

In recent years, substantial progress has been accomplished regarding the employment of CWRs for expanding the genetic resources towards improvement of agronomically important crops in the context of the ongoing climate change and the ever-increasing world population. Adverse environmental conditions compromise the yield and quality of important crops and may severely challenge food security worldwide. In crops of high economic value, such as legumes, tomato, and woody perennials, described in this review, investigations have focused on the characterization of existing wild relatives at the morho-physiological and molecular level under a variety of abiotic stress conditions. On many occasions, the genetic basis of abiotic stress tolerance was explored by comparative genomic and transcriptomic analyses between wild relatives and the cultivated species revealing genomic regions or specific stress-responsive genes and gene networks associated with successful survival under stresses, such as drought, heat, cold, and salt stress. The outcomes of these studies will be highly valuable for the development and screening of improved genotypes in breeding programs and ultimately will result in varieties with advantageous traits that impart climate-resilience. 

Nevertheless, more extensive studies should be undertaken, and further use of wild relatives should be sought out. Despite their high value as a plant genetic resource and their multiple uses in plant breeding, most CWRs are endangered or close to extinction. Furthermore, about 70% of them require immediate collection and conservation in gene banks, and 95% of them are under-represented in existing collections [177]. This calls for concerted actions at national, regional, and international levels for prioritization and systematic conservation of this important natural resource. 

Notably, much effort has been undertaken across countries to generate prioritized inventories for crop wild relatives (annual and perennial plants). These aim at proper assessment and efficient conservation, both in situ (land protection) and ex situ (seed banks), of unexplored or underexploited wild genetic resources [9,178,179,180,181]. Breeding programs focusing on introgression of wild genetic material into cultivated crops will result in climate-resilient varieties with low-input requirements. Exploiting the full potential of CWRs for developing well-adapted, climate-smart varieties that maintain high-quality produce is in line with the EU (European Union) Green Deal objectives and UN (United Nations) SDG (sustainable development goals) and ultimately will contribute greatly to sustainable agricultural production.

## Figures and Tables

**Table 1 plants-12-00328-t001:** Crop wild relatives (CWRs) of the respective crop genera that present tolerance to abiotic stresses.

Species	Type of Tolerance	Wild Species	Source
Tomato	drought tolerance	*Solanum habrochaites* (syn. *Solanum hirsutum)*	[20]
		*S. pennellii*	[21]
		*S. pimpinellifolium*	[22,23]
		*S. cheesmanii*	[24]
		*S. chilense*	[25]
		*Solanum sitiens*	[26]
	salt tolerance	*S. pennellii*	[27,28]
		*S. pimpinellifolium*	[27]
		*S. hirsutum* (syn. *S. habrochaites*)	[29]
		*Solanum parviflorum*	[30]
	heat tolerance	*S. habrochaites* (syn. *S. hirsutum)*	[20]
		*S. pennellii*	[31]
		*S. pimpinellifolium*	[31]
		*S. cheesmanii*	[32]
		*Solanum chmielewskii*	[33]
Alfalfa	drought, salt, cold tolerance	*Medicago truncatula*	[34]
		*Medicago ruthenica*	[35,36]
		*Medicago polymorpha*	[37]
		*Medicago falcata*	[38]
Cowpea	drought tolerance	*Vigna exilis*	[39]
		*Vigna heterophylla*	[40]
		*Vigna kirkii*	[40]
		*Vigna trilobata*	[39]
		*Vigna riukiensis*	[39]
	heat tolerance	*Vigna hainiana*	[40]
		*Vigna stipulacea*	[40]
	salinity tolerance	*Vigna luteola*	[40]
		*Vigna marina*	[41,42]
		*Vigna nakashimae*	[43]
		*Vigna riukiuensis*	[43,44]
		*Vigna trilobata*	[43,44]
		*Vigna vexillata*	[40]
		*Vigna trilobata*	[40]
	extreme types of soils	*Vigna minima*	[45]
		*Vigna indica*	[46]
	water-logging tolerance	*Vigna vexillata*	[47]
Groundnut	drought tolerance	*Arachis dardani*	[48]
		*Arachis diogoi*	[49]
		*Arachis duranensis*	[50,51]
		*Arachis glabrata*	[52]
		*Arachis magna*	[53]
	heat tolerance	*Arachis diogoi*	[49]
		*Arachis duranensis*	[54]
		*Arachis glabrata*	[55]
		*Arachis ipaensis*	[54]
	cold tolerance	*Arachis duranensis*	[56]
		*Arachis glabrata*	[52]
		*Arachis paraguariensis*	[55]
	salinity tolerance	*Arachis diogoi*	[49]
		*Arachis duranensis*	[51,56]
		*Arachis glabrata*	[52]
	UV-exposure tolerance	*Arachis stenosperma*	[57]
Apple	drought tolerance	*Malus prunifolia*	[58]
		*Malus sieversii*	[59,60]
	heat tolerance	*Malus prunifolia*	[61]
		*Malus sieversii*	[60]
	cold tolerance	*Malus prunifolia*	[59,60]
		*Malus baccata*	[62]
		*Malus sieversii*	[60]
Cranberry	cold tolerance	*Vaccinium oxycoccos*	[63]
Grapevine	drought tolerance	*Vitis yeshanensis*	[64]
	salt tolerance	*Vitis sylvestris*	[65]

## Data Availability

Not applicable.

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
