# Peer review of "Crop Wild Relatives: A Valuable Source of Tolerance to Various Abiotic Stresses"

_plants, 2023, doi:10.3390/plants12020328_

Round 1

Reviewer 1 Report

This article discussed about crop wild relatives as a source against environmental stresses which is a beneficial addition to literature and future research studies. Before recommending this article for publication, there are some shortcomings for that should be resolve.

Line 15 specify the “certain crops”

Line 15 specify crop traits which were considered in this study.

Methodology of data collection should be mention in the abstract.

Also mention specific main findings of the study in the abstract.

Abstract looks very general in the current way.

Line 53 in introduction must be cited.

Line 55 also lack reference. The following studies could be cited.

https://doi.org/10.3390/genes13101699,

https://doi.org/10.3390/life12111922,

section numbers must be in sequence.

Line 227 the first sentence looks like conclusion or finding. I mean the section is started very directly. Here the authors should provide some background and then present the findings.

The authors are advised to present genome size and important functional genes of the considered species in every section. It can further strengthen the conclusion of the study.

The mechanism against considered stresses of the wild relatives should also be included.

Author Response

Thank you very much for your fruitful comments. All suggestions were taken in to account and corrected in the manuscript.

Reviewer 2 Report

Dear Authors and Editors,

Wild species are still underutilized for the development of new adaptive plant varieties. This is necessary in order to expand the gene pool of cultivated plants and their ability to survive extreme conditions, without the intervention of expensive technologies and chemicals. The authors present concisely but in detail the possibilities and achievements of interspecific hybridization of important plants. I think this work will be useful and will give new ideas to breeders trying to create adaptive cultivars.

A few places that should be clarified:

Line 221 Sentence must be corrected: Twenty-one and 10ten drought-responsive TFs from the MYB/MYB-related family

Lines 234-235 I suggest to chage the sentence: We think MrERF and 234 MrbZIP can increase the germination rate of Medicago under salt stress because they greatly increased 235 the germination rate of transgenic tobacco lines.

Lines 394, 402, 476 reference boxes in bold

Lines 592, 660, 674 to check and supplement the references

Line 647 My suggestion to supplement the reference: Pratap, A., Kumar, J. (2014). Alien Gene Transfer in Crop Plants: An Introduction. In: Pratap, A., Kumar, J. (eds) Alien Gene Transfer in Crop Plants, Volume 1. Springer, New York, NY. https://doi.org/10.1007/978-1-4614-8585-8_1

Line 654 My suggestion to supplement the reference: Padulosi, S. & Ng, N. Q. (1997). Origin, taxonomy, and morphology of Vigna unguiculata (L.) Walp. In B.B. Singh, D.R. Mohan Raji and K.E. Dashiel, Advances in cowpea research. Ibadan, Nigeria: IITA, (p. 1-12).

Author Response

Thank you very much for your fruitful comments and suggestions. Every comment has been answered and additional references have been added to the manuscript.
